# EDSNN: EDGE DETECTION WITH SPIKING NEURON NETWORK

## ABSTRACT

Edge detection has made great progress under the development of Artificial Neural Networks (ANNs), particularly Convolutional Neural Networks (CNNs) and Transformers, some of them even have achieved a beyond human-level performance. However, these methods come with complex designs and high energy consumption. Spiking Neural Networks (SNNs), with their low energy consumption and biological interpretability, offer a promising solution to address these issues. In this work, we propose the first SNN-based method named EDSNN (Edge Detection with Spiking Neural Network) for edge detection. We construct a novel Spiking Multi-Scale Block (SMSB) to effectively utilize multi-scale information, thereby helping the network generate precise and clean edge maps. In addition, to more accurately decode spike trains, we present a Membrane Average Decoding (MAD) method in the prediction block. Our method has the advantages of remarkable efficiency and high performance across multiple datasets. It surpasses the human-level performance on BSDS500 (ODS=0.804 vs. ODS=0.803) while consuming only 14.64 mJ, remains competitive performance among top-performing ANN-based approaches on NYUDv2 (ODS=0.750), and achieves state-of-the-art performance on BIPED (ODS=0.891). Our codes are publicly available in supplementary materials.

## 1 INTRODUCTION

Edge detection is a longstanding and fundamental task in computer vision, aiming to identify semantically meaningful object boundaries Deng et al. (2018). It is crucial in many high-level tasks such as object detection Zhou et al. (2022a); Yao & Wang (2023) and image segmentation Wang et al. (2022); Jin et al. (2023). In recent years, edge detection has attracted significant attention from researchers and has achieved remarkable progress. Some state-of-the-art (SOTA) methods based on Artificial Neural Networks (ANNs) have achieved a beyond human-level performance Liu et al. (2024). However, these high-performing edge detection methods are usually associated with complex designs and extremely high energy consumption Zhou et al. (2024a). We believe that as a foundational task, edge detection needs simpler and more energy-efficient solutions.

Spiking neural networks (SNNs) offer a potential solution to this problem. As the third generation of neural networks, SNNs compute and transmit information using spike signals Maass (1997). The characteristics of their binary which involves all-or-nothing computations, make them particularly well-suited for binary classification tasks like edge detection. Moreover, because SNNs transmit and compute information solely through spikes, the operations within the network are limited to addition, leading to a significant reduction in energy consumption. With these advantages, SNNs have demonstrated excellent performance in some high-level computer vision tasks such as image classification Zhou et al. (2024b; 2022b), object detection Fan et al. (2024), and depth estimation Rançon et al. (2022). However, the effective application of SNNs to edge detection remains a topic that requires further exploration.

To fill this gap, we propose an SNN-based edge detection method called EDSNN, marking the first attempt of the SNNs for edge detection. Specifically, the EDSNN employs an encoder-decoder architecture, which facilitates the preservation and utilization of high-resolution spatial information throughout the network. For the encoder, we convert the VGG network Simonyan & Zisserman (2014) into an SNN version, enabling multi-scale feature extraction in the spiking domain.

Then, in edge detection, precisely locating edge pixels and generating clean edge maps have long been major challenges Deng et al. (2018). To address this issue, we propose the Spiking Multi-Scale Block (SMSB) in the decoder. The SMSB employs parallel convolutions with different kernel sizes and dilation rates to capture multi-scale features. Such a strategy can integrate local precision with long-range context, enabling the network to consider both detailed edge characteristics and their surrounding noise, thereby helping to comprehensively understand edge characteristics across different scales, consequently enhancing noise suppression. Subsequently, utilizing the proposed SMSB and Nearest-Neighbor Interpolation, we perform spike-friendly upsampling operations Rançon et al. (2022) to restore the features to the original resolution.

Furthermore, in SNNs, both computation and information transmission are carried out using spike trains. Consequently, efficient and accurate decoding of these spike trains at the final output stage has become a crucial issue in the SNN field. Currently, common decoding methods include Spiking Rate Decoding (SRD), Spiking Count Decoding (SCD), and Last Membrane Potential Decoding (LMPD). While Fan et al. (2024) have demonstrated that SRD is more conducive to convergence than SCD, we argue that SRD merely decodes spike trains into a few fixed discrete values, potentially limiting the model's expressive power. On the other hand, LMPD suppresses spike firing and uses the final accumulated membrane potential for decoding, which may diminish the impact of earlier time steps. To address the limitations of these existing decoding approaches, we present the Membrane Average Decoding (MAD) method. The MAD modifies the last layer of neurons to apply a decay function to the membrane synaptic inputs and then accumulate them over time, ultimately outputting the accumulated membrane potential. This process essentially averages the membrane synaptic inputs, hence the name Membrane Average Decoding. Finally, we accurately predict the edge map at the final stage of our model using the proposed MAD method.

The main contributions of this work can be summarized as follows:

- To enable the model to generate clean edge maps in a spike-friendly manner, we propose the Spiking Multi-Scale Block (SMSB). By integrating convolutions with varying receptive fields, this block can suppress the false positive edge pixels and improve the accuracy of true edge location.
- To enhance the efficiency of spike decoding, we propose the Membrane Average Decoding (MAD) method. This approach not only improves the model's expressive ability but also fully considers information from all time steps.
- We propose EDSNN, the first SNN-based edge detection method, which adopts a simple encoder-decoder network architecture. Extensive experiments demonstrate the remarkable performance of our method. Specifically, it surpasses human-level performance on BSDS500 and achieves SOTA performance on BIPED. All the experiment results suggest that SNNs have a promising potential for edge detection.

## 2 RELATED WORK

### 2.1 EDGE DETECTION

Edge detection is a fundamental research task in computer vision that has seen significant developments over the years. Early methods primarily rely on calculating image derivatives information which includes Sobel Sobel (1970), Laplacian Jain et al. (1995), and Canny Canny (1986). The advent of Artificial Neural Networks (ANNs), especially Convolutional Neural Networks (CNNs) and Transformers, has brought about revolutionary progress in this field, giving rise to many ANN-based methods. HED Xie & Tu (2015) utilizes a fully convolutional neural network to perform end-to-end edge detection. RCF Liu et al. (2017) exploits multi-scale and multi-level features to enhance edge detection performance. BDCN He et al. (2019) introduces a bi-directional cascade structure to refine edge predictions progressively. DexiNed Poma et al. (2020) employs a dense extreme inception architecture for improved edge localization. PiDiNet Su et al. (2021) utilizes pixel difference convolutions for high-efficiency edge detection. EDTER Pu et al. (2022) proposes a two-stage Transformer-based architecture for accurate edge detection. As for precise edge detection, some novel loss functions are proposed, such as LPCB Deng et al. (2018) and DSCD Deng & Liu (2020). Recent methods focus on exploring the uncertainty arising from multi-annotators in datasets, including UAED Zhou et al. (2023), RankED Cetinkaya et al. (2024), and BetaNet Li et al. (2023). These methods provide a new perspective on edge detection. However, ANN-based

methods with high computational demands translate to significant energy consumption, which can be problematic for deployment on edge devices or in energy-sensitive scenarios.

## 2.2 Spiking Neural Network

SNNs are biologically inspired models that process information through discrete spikes, offering advantages in energy efficiency and temporal data processing over traditional ANNs. To realize these advantages of SNNs, numerous researchers have proposed various SNN neuron models, such as the Hodgkin-Huxley (H-H) model Hodgkin & Huxley (1952), the Izhikevich model Izhikevich (2003), the Integrate-and-Fire (IF) model Burkitt (2006), the Leaky Integrate-and-Fire (LIF) model Abbott (1999), and the Parametric Leaky Integrate-and-Fire (PLIF) model Fang et al. (2021). In this work, we adopt the IF model due to its balance of computational efficiency and ability to capture essential neuronal dynamics.

Currently, most SNN-based approaches are primarily employed to address various vision problems, including object detection, image classification, and image segmentation. SFOD Fan et al. (2024) proposes a novel multi-scale feature fusion and optimized spiking decoding strategies for high-performance object detection. Meta-SpikeFormer Yao et al. (2024) combines convolution and Transformer blocks with spike-driven self-attention, which achieves state-of-the-art performance on top-1 accuracy classification. Spiking U-Net Li et al. (2024) achieves comparable accuracy to traditional CNNs while consuming significantly less energy, demonstrating the potential of neuromorphic computing for efficient image processing. These researches fully demonstrate the potential of SNNs in computer vision. However, research on SNNs in edge detection remains notably limited. Given that edge detection is a task highly analogous to semantic segmentation, we believe that SNNs could be equally effective in this field.

## 3 Method

### 3.1 Overview

The whole architecture of our EDSNN is shown in Fig. 1. We adopt a simple encoder-decoder structure to construct the EDSNN and such a simple structure has been proven to have strong capabilities in ANNs Liu et al. (2024); Deng et al. (2018). First, we use the direct coding method to convert static RGB images into spike trains. Unlike the rate coding method, which combines fixed probability model sampling, direct coding introduces a learnable encoding layer Kim et al. (2022). After passing through this encoding layer, the output is repeated T times and fed into IF neurons, generating spike trains. This enhances the learning capability of the model. To further optimize the encoding process, we improve it by directly repeating the RGB image T times and then feeding it into a Conv7×7-tdBN-IF structure to generate spike trains. By introducing tdBN, the gradient propagation within the encoding layer is effectively stabilized Zheng et al. (2021).

After the coding process, the generated spike trains are fed into the encoder for multi-scale feature extraction. The extracted features at different levels are then passed through the Skip Module to further refine feature representations at the same resolution. Specifically, the Level 1 encoder features are processed by a Conv3×3-tdBN-IF block, Level 2-5 features are processed by a Conv1×1-tdBN-IF block, and Level 6 features are processed by two sequential Spiking Resblocks Zheng et al. (2021). Then, the Level 6 features from the Skip Module are sent to the decoder as the primary features for upsampling. The output from the other level of Skip Modules is used as supplementary information and is combined with the corresponding decoder features through membrane-based addition.

Finally, during the training process, we use n×Up Prediction Blocks to restore the decoder output of each layer to the original resolution, enabling deep supervision. During inference, we take the output from Level 1 as the final edge map. Notably, in the n×Up Prediction Block, we adopt the Membrane Average Decoding (MAD) method to decode information across multiple time steps.

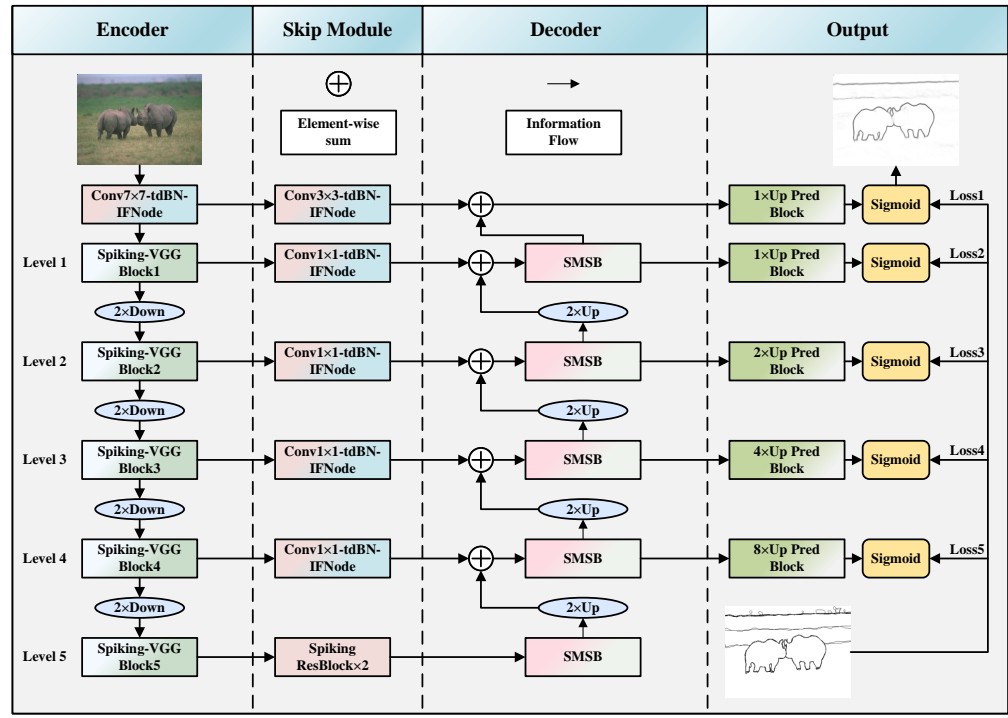

Figure 1: The architecture of our EDSNN. It consists of the encoder, skip module and decoder. The encoder uses Spiking-VGG blocks to extract hierarchical features, while the skip module refines the features. The decoder employs SMSB for multi-scale feature fusion. Our EDSNN produces edge maps at multiple scales through n×Up Prediction Blocks and Sigmoid activations, combining low-level and high-level features for accurate edge location.

## 3.2 ENCODER

As the first attempt to apply SNNs to edge detection and to facilitate comparisons with other networks, we propose the Spiking VGG network, based on the widely used VGG network Simonyan & Zisserman (2014) in ANNs for edge detection Liu et al. (2019); Xie & Tu (2015), and employ it as the encoder. Additionally, in ANNs, large-scale pre-trained networks on ImageNet are typically used as encoders to enhance feature extraction capabilities for edge detection Liu et al. (2024). However, we believe that such large-scale pre-trained weights not only waste resources but also against the simplicity and energy efficiency that should be prioritized for foundational tasks like edge detection. Therefore, we do not pre-train our encoder.

The vanilla VGG network is a deep convolution neural network composed of multiple stacked small convolution layers. To convert it into a spiking version, we replace its activation functions with IF neurons, enabling computation and information transmission through spikes. Additionally, tdBN is introduced between the convolution layers, IF neurons are employed to stabilize gradient propagation and accelerate convergence.

## 3.3 DECODER

Following the multi-scale feature learning process, the network needs to upsample these features to enhance edge pixel location accuracy and align the final output dimensions with the edge map. However, Bilinear Interpolation, commonly used in ANNs, is unsuitable for SNNs due to its reliance on multiplication and division operations Rançon et al. (2022). Moreover, while transposed convolution (deconvolution) is spike-friendly Fan et al. (2024), it often introduces checkerboard artifacts

Odena et al. (2016). To overcome these limitations, we employ Nearest-Neighbor Interpolation for the upsampling process.

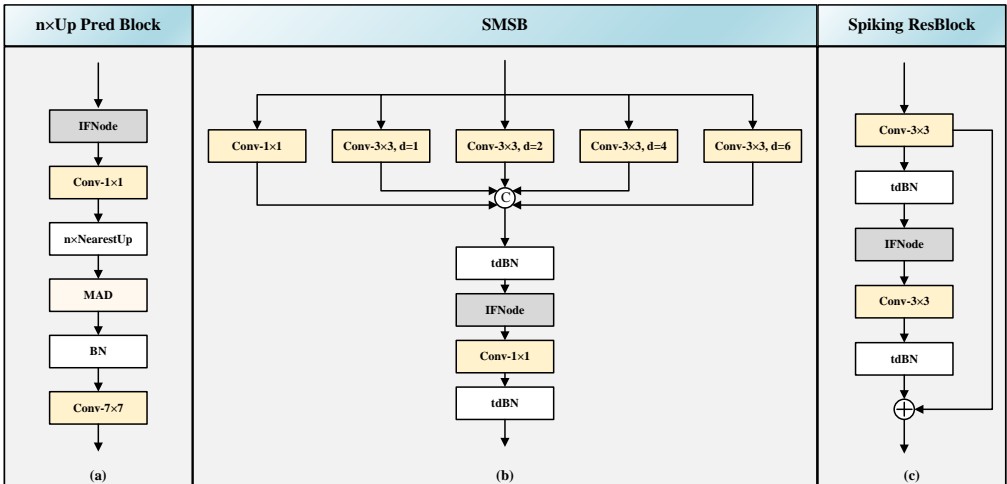

Figure 2: The architecture of n×Up Prediction Block, SMSB, and Spiking ResBlock, respectively.

In the decoder, there is a need to aggregate information from the Skip Module and the higher-level upsampled features within the decoder. Consequently, finding out how to optimally fuse these features in each block of the decoder becomes a significant challenge. Furthermore, generating precise edge maps has long been a focal point in edge detection. In our view, the coarseness of edge inference results stems from an abundance of edge artifacts (false positives) surrounding true edges (true positives). These artifacts significantly interfere with the inference of true edges, inevitably leading to thick edge predictions.

To address these issues, we propose the **Spiking Multi-Scale Block** (SMSB), as illustrated in Fig. 2 (b). Within the SMSB, we integrate five convolutions with various kernel sizes and dilation rates. These multi-scale features are then aggregated through concatenation and 1×1 convolution. In edge detection, true edge pixels are typically associated with objects or structures, while noise edges lack semantic coherence. Therefore, the SMSB can effectively integrate local precise spatial cues from smaller receptive fields with long-range context information from larger receptive fields. This integration is crucial because it can provide structural information to enhance the network's ability to distinguish true edge pixels from noise pixels, resulting in the production of refined edge maps.

## 3.4 MEMBRANE AVERAGE DECODING

Currently, the most commonly used spiking decoding methods are Spiking Rate Decoding (SRD) and Last Membrane Potential Decoding (LMPD). Specifically, SRD involves summing the emitted spikes from the last layer of spiking neurons and dividing by the number of time steps T to calculate the spiking rate, which represents the decoding result of the model. While this method preserves the spiking characteristics of the network, it significantly compromises its expressive capacity. This is because it only represents the output of the model as a few fixed discrete values, diminishing the expressive power of the model. LMPD suppresses spike firing in the final layer and uses the accumulated membrane potential as the final output. Although this method enhances the expressive capacity of the model, it disregards the influence of earlier time steps. Furthermore, the continuous accumulation of membrane potential for output can lead to an amplification of subtle edge artifacts, which is harmful to generating clean edge maps.

To address these issues, we propose the Membrane Average Decoding (MAD) method. This method modifies the neurons in the final layer as follows:

$$\begin{cases} V_t = V_{t-1} + \dfrac{1}{T}X_t \\ O = V_T \end{cases} \tag{1}$$

where $V_t$ and $X_t$ represent the membrane potential of the neuron and the input at time step t, respectively, O denotes the neuron output, and T indicates the total number of time steps.

Essentially, this method averages the membrane synaptic input at each time step, which not only considers information from all time steps but also overcomes the issue of inadequate expressive capacity in SRD. Moreover, this approach offers computational simplicity during inference, as it only requires dividing the parameters of the previous layer by T. This allows for addition-only operations during decoding, thereby eliminating division operations.

### 3.5 HYBRID FOCAL LOSS

We employ the hybrid focal loss function which is formulated in this work Liu et al. (2024) to supervise the training process. The hybrid focal loss function consists of focal tversky loss and focal loss, which can be defined as follows:

$$\begin{aligned} L_{HFL} &= L_{FT} + \lambda \cdot L_{FL} \\ &= \left( \frac{\sum_{i=1}^{N} p_i g_i + (1-\beta) \sum_{i=1}^{N} (p_i(1-g_i))^2 + \beta \sum_{i=1}^{N} ((1-p_i)g_i)^2 + C}{\sum_{i=1}^{N} p_i g_i + C} \right)^{\gamma} \\ &\quad - \lambda \cdot \omega \sum_{i=1}^{N} \left[ (1-p_i)^2 g_i \log p_i + p_i^2 (1-g_i) \log (1-p_i) \right] \end{aligned} \tag{2}$$

where $p_i$ and $g_i$ represent the value of $i$-th pixel in a predicted edge map and its corresponding label image, respectively. $p_i(1-g_i)$ and $(1-p_i)g_i$ represent false positive pixels (FPs) and false negative pixels (FNs). $\gamma = 0.75$ represents the focusing parameter and $C = 1 \times 10^{-7}$ is a constant number to prevent the numerator/denominator from being 0. $(1-\beta)$ and $\beta$ are parameters to balance the weights between FPs and FNs. $N$ represents the total number of pixels in an image. $(1-p_i)^2$ is the modulating factor, and $\omega = 0.25$ is the balance factor for positive and negative pixels. We optimize the performance by adjusting hyper-parameters in the loss function. The specific experiment results are provided in the supplementary materials.

## 4 EXPERIMENTS

### 4.1 DATASETS AND IMPLEMENTATION

**Datasets.** We select three widely used datasets to evaluate the performance of our EDSNN: BSDS500 Arbelaez et al. (2010), NYUDv2 Silberman et al. (2012), and BIPED Poma et al. (2020). BSDS500 comprises 200 training images, 100 validation images, and 200 test images. Each image is annotated by multiple annotators (around 5 to 7). Following previous works Deng et al. (2018), we also incorporate the PASCAL VOC Context dataset Mottaghi et al. (2014), containing 10103 images, as additional training data to further enhance the model's performance. NYUDv2 dataset contains 1449 pairs of images, each pair comprising an RGB image and its corresponding depth map. These image pairs are split into 381 for training, 414 for validation, and 654 for testing. BIPED is a high-quality dataset with 250 high-resolution images (1280×720) captured in outdoor scenes. All images are divided into a training set of 200 images and a test set of 50 images. During the training phase, we merge the training and validation images of BSDS500 into a single subset. The same procedure is applied to the NYUDv2. For BIPED, we adopt their default configuration. As for data augmentation, we follow the protocols established in previous works Deng et al. (2018). The strategy involves first applying three-way flipping to the images (horizontal, vertical, and both), followed by rotating each flipped image through 24 different angles. This data augmentation strategy is consistently applied across all three datasets.

**Implementation Details.** We adopt the SpikingJelly Fang et al. (2023) deep-learning framework to implement our network. Specifically, we set the mini-batch size to 8 in BSDS500 and 4 in NYUDv2, respectively. We randomly crop the images in BIPED to $320 \times 320$ for training, as the original resolution is relatively large, and the mini-batch size is set to 8. The initial learning rate is set to $1 \times 10^{-4}$ and the learning rate decay is 0.1. We decay the learning rate every 3 epochs and adopt the Adam for optimization. The number of total training epochs is set to 21. All the experiments are performed using a single Tesla A40 GPU.

**Evaluation Metrics.** To evaluate the performance of EDSNN, we employ widely adopted metrics in edge detection, including ODS (Optimal Dataset Scale), OIS (Optimal Image Scale), and AP (Average Precision). Before metric computation, we process the predicted edge maps using non-maximum suppression and morphological thinning. During the evaluation phase, the localization tolerance is set to 0.0075 for BSDS500 and BIPED, while for NYUDv2, it is set to 0.011.

Additionally, we report the energy consumption to quantify the energy efficiency of the network, which is frequently utilized in Spiking Neural Networks. SNN energy efficiency stems from performing accumulation calculations (AC) only when neurons fire. However, many current SNN works cannot ensure full-spiking networks, so we consider both AC and multiplication-addition (MAC) operations when calculating energy consumption. For ANNs, we focus on MAC operations, as they dominate. Following previous studies Qu et al. (2024); Kim et al. (2020); Fan et al. (2024), we use $E_{MAC} = 4.6\text{pJ (FLOAT32)}/3.2\text{pJ (INT)}$, $E_{AC} = 0.9\text{pJ (FLOAT32)}/0.1\text{pJ (INT)}$. The energy consumption formulas for SNNs and ANNs are as follows, with fr representing firing rate, T representing time steps, and $\eta$ representing the number of operations.

$$E_{SNNs} = T \times fr \times (E_{AC} \times \eta_{AC} + E_{MAC} \times \eta_{MAC}) \tag{3}$$

$$E_{ANNs} = T \times E_{MAC} \times \eta_{MAC} \tag{4}$$

## 4.2 ABLATION STUDY

Table 1: The results of ablation study. All the backbones are without any large-scale pre-trained weights.

| Backbone | Decode Method | Decoder Block | T | Energy (mJ) | Firing Rate (%) | ODS | OIS | AP |
|---|---|---|---|---|---|---|---|---|
| VGG16 | MAD | Conv3×3 | 2 | 6.88 | 16.74 | 0.779 | 0.796 | 0.782 |
| **VGG16** | **MAD** | **SMSB** | **2** | **7.59** | **12.53** | **0.783** | **0.802** | **0.785** |
| VGG16 | SRD | SMSB | 2 | 12.31 | 20.32 | 0.779 | 0.798 | 0.775 |
| VGG16 | LMPD | SMSB | 2 | - | 12.10 | 0.780 | 0.798 | 0.782 |
| **VGG16** | **MAD** | **SMSB** | **2** | **7.59** | **12.53** | **0.783** | **0.802** | **0.785** |
| VGG13 | MAD | SMSB | 2 | 7.15 | 12.15 | 0.780 | 0.798 | 0.783 |
| **VGG16** | **MAD** | **SMSB** | **2** | **7.59** | **12.53** | **0.783** | **0.802** | **0.785** |
| VGG19 | MAD | SMSB | 2 | 7.83 | 12.58 | 0.778 | 0.798 | 0.782 |
| VGG16 | MAD | SMSB | 1 | 3.46 | 11.44 | 0.776 | 0.795 | 0.780 |
| VGG16 | MAD | SMSB | 2 | 7.56 | 12.53 | 0.783 | 0.802 | 0.785 |
| VGG16 | MAD | SMSB | 3 | 13.20 | 14.53 | 0.785 | 0.802 | 0.788 |
| **VGG16** | **MAD** | **SMSB** | **4** | **18.19** | **15.02** | **0.785** | **0.804** | **0.788** |
| VGG16 | MAD | SMSB | 5 | 15.32 | 10.12 | 0.782 | 0.803 | 0.785 |

**The Effectiveness of SMSB:** We validate the effectiveness of SMSB, as demonstrated in rows 1 and 2 of Table 1. When replacing SMSB with standard 3×3 convolution in the decoder, we observed a slight decrease in energy consumption. However, this substitution led to a significant drop in the ODS, OIS, and AP scores. These results strongly support the effectiveness of our proposed SMSB, demonstrating that this block can achieve better edge location capability while maintaining a similar level of energy consumption.

**The Effectiveness of MAD:** We compared our proposed MAD with other commonly used methods, specifically SRD and LMPD. The comparative results are presented in rows 3-5 of Table 1. The MAD method not only maintains comparable energy consumption but also achieves higher ODS,

OIS, and AP scores compared to these alternatives. These findings validate our analysis presented in Section 3.4.

**Depth of Spiking VGG:** We also investigate the impact of depth on the Spiking VGG architecture, with results presented in rows 6-8 of Table 1. Our findings indicate that model performance improves as the depth increases from 13 to 16 layers. However, further increases in depth lead to overfitting, suggesting that Spiking VGG16 is the optimal configuration for this task.

**Size of Time Steps:** As shown in rows 9-13 of Table 1, we investigate the impact of different time steps on model performance. It can be observed that in the Edge Detection task, model performance gradually improves as the number of time steps increases. However, unlike in object detection tasks Su et al. (2023), performance begins to decline after T increases beyond 4. We attribute this phenomenon to the increased model capacity as T grows, which easily leads to overfitting for low-level tasks like Edge Detection.

## 4.3 COMPARISON WITH STATE-OF-THE-ARTS

**BSDS500:** We compare our EDSNN with some traditional edge detectors such as Canny Canny (1986), gPb-UCM Arbelaez et al. (2010), SCG Ren & Bo (2012), PMI Isola et al. (2014), SE Dollár & Zitnick (2014), OEF Hallman & Fowlkes (2015) and MES Sironi et al. (2015), and CNN-based methods such as DeepEdge Bertasius et al. (2015), DeepContour Shen et al. (2015), HED Xie & Tu (2015), AMH-Net Xu et al. (2017), RCF Liu et al. (2017), CED Wang et al. (2017), LPCB Deng et al. (2018), BDCN He et al. (2019), DexiNed Poma et al. (2020), DSCD Deng & Liu (2020), PiDiNet Su et al. (2021), UAED Zhou et al. (2023), and RankED Cetinkaya et al. (2024), and Transformer-based method such as EDTER Pu et al. (2022). The results are summarized in Table 2 and some examples are shown in Fig. 3.

As shown in Table 2, our EDSNN achieves remarkable performance with ODS, OIS, and AP scores of 0.804, 0.825, and 0.823 respectively. Significantly, our EDSNN is the first SNN-based method that has outperformed human performance in this task (0.804 vs. 0.803). While the EDTER shows the highest performance (with EDTER†‡ achieving 0.848, 0.865, and 0.903 for ODS, OIS, and AP), this comes at a substantial energy consumption of 3054.4 mJ, which over 200 times more than ours (14.64 mJ). Similarly, top-performing CNN-based methods such as UAED and RankED, despite their high accuracy, consume significantly more energy (669.57 mJ and 1600.62 mJ respectively) compared to EDSNN. All these results fully demonstrate our EDSNN can achieve a remarkable balance between high performance and energy consumption. It is noteworthy to emphasize that EDSNN achieves comparable performance to CNN-based methods (such as HED and CED) without any pre-trained weights, while maintaining a relatively lower energy consumption. Additionally, as evidenced in Fig. 3, our method can generate clean and refined contour maps. This enhanced level of detail further elucidates the substantial potential of Spiking Neural Networks (SNNs) in edge detection.

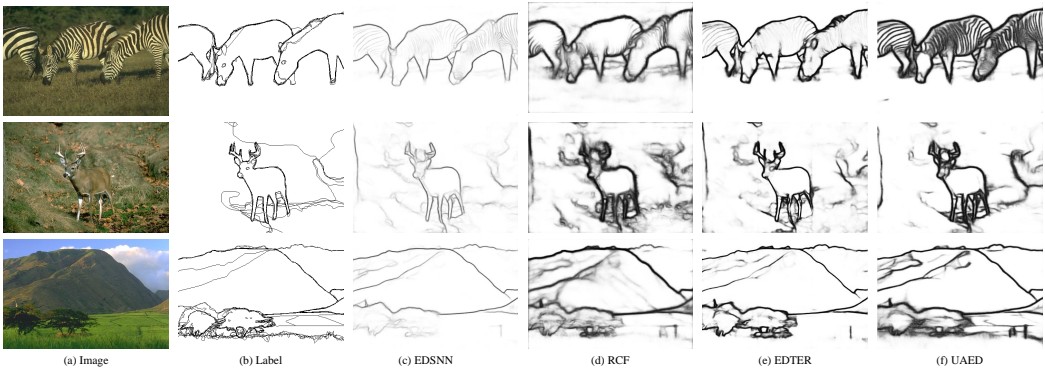

Figure 3: Some examples from different methods on BSDS500. From (a) to (f): (a) are the raw images, (b) are their corresponding label images, (c) to (f) are the predictions of our EDSNN, RCF, EDTER, and UAED, respectively.

Table 2: Quantitative comparison results on BSDS500 dataset. † indicates using extra PASCAL VOC Context dataset in the training process. ‡ indicates the multi-scale testing. Energy means energy consumption.

| Methods | | ODS | OIS | AP | Energy(mJ) |
|---|---|---|---|---|---|
| Traditional | Canny | 0.611 | 0.676 | 0.520 | - |
| | gPb-UCM | 0.729 | 0.755 | 0.745 | - |
| | SCG | 0.739 | 0.758 | 0.773 | - |
| | PMI | 0.741 | 0.769 | 0.799 | - |
| | SE | 0.743 | 0.764 | 0.800 | - |
| | OEF | 0.746 | 0.770 | 0.820 | - |
| | MES | 0.756 | 0.776 | 0.756 | - |
| CNN-based | DeepEdge | 0.753 | 0.772 | 0.807 | - |
| | DeepContour | 0.757 | 0.776 | 0.790 | - |
| | HED | 0.788 | 0.808 | 0.840 | 604.66 |
| | AMH-Net | 0.798 | 0.829 | 0.869 | - |
| | CED | 0.794 | 0.811 | - | - |
| | RCF | 0.798 | 0.815 | - | 551.56 |
| | LPCB | 0.800 | 0.816 | 0.808 | 590 |
| | BDCN | 0.806 | 0.826 | 0.847 | 1542.29 |
| | PiDiNet | 0.789 | 0.803 | - | 72.02 |
| | DexiNed | 0.729 | 0.745 | 0.583 | 710.69 |
| | DSCD | 0.802 | 0.817 | - | 717.37 |
| | UAED | 0.829 | 0.847 | 0.892 | 669.57 |
| | RankED | 0.824 | 0.840 | 0.895 | 1600.62 |
| Transformer-based | EDTER | 0.824 | 0.841 | 0.880 | 3054.4 |
| | EDTER† | 0.832 | 0.847 | 0.886 | |
| | EDTER‡ | 0.840 | 0.858 | 0.896 | |
| | EDTER†‡ | 0.848 | 0.865 | 0.903 | |
| SNN-based | EDSNN | 0.785 | 0.804 | 0.788 | 18.19 |
| | EDSNN† | 0.798 | 0.818 | 0.804 | **14.64** |
| | EDSNN†‡ | **0.804** | **0.825** | **0.823** | |

**NYUDv2:** In NYUDv2, we perform experiments on three different types of images: RGB, HHA, and RGB-HHA. The RGB-HHA means directly averaging the predictions from RGB and HHA. We compare our EDSNN against the SOTA ANN-based methods including HED Xie & Tu (2015), RCF Liu et al. (2017), AMH-Net Xu et al. (2017), LPCB Deng et al. (2018), BDCN He et al. (2019), PiDiNet Su et al. (2021), RankED Cetinkaya et al. (2024), and EDTER Pu et al. (2022). The results are presented in Table 3. As evidenced in Table 3, EDSNN demonstrates performance comparable to ANN-based methods, aligning with the results observed on BSDS500. The consistent performance across datasets (NYUDv2 and BSDS500) underscores our method's robustness and transferability. Specifically, EDSNN achieves highly competitive performance on the RGB-HHA data, with ODS=0.750, OIS=0.766, and AP=0.767, while maintaining lower energy consumption. These comparison results substantiate the robust potential and generalization capabilities of SNNs in edge detection.

**BIPED:** We adopt six ANN-based methods for comparison which consist of SED Akbarinia & Parraga (2018), HED Xie & Tu (2015), RCF Liu et al. (2017), BDCN He et al. (2019), DexiNed Poma et al. (2020), CATS Huan et al. (2021), and CED-ADM Li & Shui (2021). All the quantitative results are listed in Table 4. The single-scale testing version of EDSNN exhibits remarkable performance, surpassing all the other ANN-based SOTA methods in the comparison. Notably, the multi-scale testing version of EDSNN (EDSNN‡) achieves the highest performance across all metrics (ODS=0.891, OIS=0.897, and AP=0.924). This performance represents a significant improvement over the second-best method, CATS, of 0.45%, 0.56%, and 13.10% in ODS, OIS, and AP respectively. This comprehensive superiority suggests that our SNN-based method effectively leverages the unique characteristics of spiking neural networks to capture intricate edge features, potentially offering a new paradigm in edge detection methodologies.

Table 3: Quantitative comparison results on NYUDv2 dataset. RGB indicates the RGB images, HHA indicates the HHA images, and RGB-HHA means averaging the predictions of RGB images and HHA images.

| Methods | RGB | | | HHA | | | RGB-HHA | | |
|---|---|---|---|---|---|---|---|---|---|
| | ODS | OIS | AP | ODS | OIS | AP | ODS | OIS | AP |
| HED | 0.720 | 0.734 | 0.734 | 0.682 | 0.695 | 0.702 | 0.746 | 0.761 | 0.786 |
| AMH-Net | 0.744 | 0.758 | 0.765 | 0.716 | 0.729 | 0.734 | 0.771 | 0.786 | 0.802 |
| RCF | 0.729 | 0.742 | - | 0.705 | 0.715 | - | 0.757 | 0.771 | - |
| LPCB | 0.739 | 0.754 | - | 0.705 | 0.715 | - | 0.762 | 0.778 | - |
| BDCN | 0.748 | 0.763 | 0.770 | 0.707 | 0.719 | 0.731 | 0.765 | 0.781 | 0.813 |
| PiDiNet | 0.733 | 0.747 | - | 0.715 | 0.728 | - | 0.756 | 0.773 | - |
| RankED | 0.780 | 0.793 | 0.826 | - | - | - | - | - | - |
| EDTER | 0.774 | 0.789 | 0.797 | 0.703 | 0.718 | 0.727 | 0.780 | 0.797 | 0.814 |
| EDSNN | **0.727** | **0.743** | **0.724** | **0.690** | **0.703** | **0.663** | **0.750** | **0.766** | **0.767** |

Table 4: Quantitative comparison results on BIPED dataset. ‡ indicates the multi-scale testing.

| Methods | ODS | OIS | AP |
|---|---|---|---|
| SED | 0.717 | 0.731 | 0.756 |
| HED | 0.829 | 0.847 | 0.869 |
| RCF | 0.843 | 0.859 | 0.882 |
| BDCN | 0.839 | 0.854 | 0.887 |
| DexiNed | 0.859 | 0.867 | 0.905 |
| CATS | 0.887 | 0.892 | 0.817 |
| CED-ADM | 0.810 | 0.835 | 0.869 |
| EDSNN | 0.888 | 0.895 | 0.920 |
| EDSNN‡ | **0.891** | **0.897** | **0.924** |

## 5 CONCLUSION

In this work, we propose the EDSNN network which is the first SNN-based method for edge detection. We build a novel Spiking Multi-Scale Block (SMSB) into the decoder to enhance its multi-scale ability, thereby suppressing the false edge pixels near the true edge pixels. This strategy can facilitate more accurate localization of edge pixels by our network. In addition, to more accurately decode spike sequences, we propose the Membrane Average Decoding (MAD) method. Our method is simple yet effective for edge detection, and the training process without relying on any large-scale pre-trained weights. EDSNN offers a highly efficient solution for edge detection, showcasing a well-balanced trade-off between energy consumption and performance. However, as a pioneering attempt at applying SNNs to edge detection, our proposed method still has room for improvement. In future work, we will explore the impact of various SNN architectures on edge detection, thereby developing more powerful SNN-based edge detection methods.

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

# A APPENDIX

In this appendix, we provide additional detailed information, including more implementation details, more ablation study about network configuration and loss function, as well as more experiment results and their visualization on BSDS500, NYUDv2, and BIPED.

## A.1 MORE IMPLEMENTATION DETAILS

**Multi-scale testing:** The process of multi-scale testing consists of three main steps: 1. Image Pyramid Construction: We create an image pyramid comprising three resolutions of the input image (0.5×, 1.0×, and 2.0×) using bilinear interpolation; 2. Multi-scale Processing: Each scaled image is independently processed through EDSNN. The resulting edge maps are then restored to the original input resolution; 3. Edge Map Fusion: The three restored edge maps are averaged to produce a final fused edge map.

## A.2 MORE ABLATION STUDY

Table 5: The results of more ablation study.

| Backbone | Stage | Loss | T | Energy (mJ) | Firing Rate (%) | ODS | OIS | AP |
|---|---|---|---|---|---|---|---|---|
| VGG16 | 4 | HFL | 2 | 5.15 | 11.00 | 0.774 | 0.794 | 0.778 |
| **VGG16** | **5** | **HFL** | **2** | **7.59** | **12.53** | **0.783** | **0.802** | **0.785** |
| VGG16 | 5 | WCE | 2 | 10.64 | 17.57 | 0.777 | 0.799 | 0.808 |
| **VGG16** | **5** | **HFL** | **2** | **7.59** | **12.53** | **0.783** | **0.802** | **0.785** |

Table 6: The results of hyperparameter for HFL.

| Backbone | $\lambda$ | $\beta$ | T | Energy (mJ) | Firing Rate (%) | ODS | OIS | AP |
|---|---|---|---|---|---|---|---|---|
| VGG16 | 0.01 | 0.7 | 2 | 9.63 | 15.91 | 0.780 | 0.799 | 0.785 |
| **VGG16** | **0.001** | **0.7** | **2** | **7.59** | **12.53** | **0.783** | **0.802** | **0.785** |
| VGG16 | 0.0001 | 0.7 | 2 | 5.33 | 8.80 | 0.776 | 0.796 | 0.786 |
| VGG16 | 0.001 | 0.6 | 2 | 7.49 | 12.37 | 0.780 | 0.797 | 0.782 |
| **VGG16** | **0.001** | **0.7** | **2** | **7.59** | **12.53** | **0.783** | **0.802** | **0.785** |
| VGG16 | 0.001 | 0.8 | 2 | 7.54 | 12.45 | 0.781 | 0.799 | 0.784 |

**Stage of Spiking VGG:** We conduct an ablation study on the configuration of stages in Spiking VGG, with results shown in rows 1 and 2 of Table 5. Although there is a slight increase in energy consumption, the configuration of the 5-stage outperforms the 4-stage in ODS, OIS, and AP. We believe that the 5-stage Spiking VGG can provide richer semantic information, thereby enhancing the model's feature extraction ability.

**Loss function for EDSNN:** We compared the HFL loss we used with WCE loss, with results shown in rows 3 and 4 of Table 5. As observed, HFL can improve the performance in edge detection with lower energy consumption. Additionally, we adjust the parameters $\lambda$ and $\beta$ in the HFL loss, as shown in Table 6, finding that the best performance is obtained at $\lambda = 0.001$ and $\beta = 0.7$.

## A.3 MORE EXPERIMENT RESULTS

We draw the Precision-Recall curves on BSDS500 and NYUDv2 which are shown in Fig. 4. The Precision-Recall curves show that our EDSNN achieves top performance on both BSDS500 and NYUDv2. Additionally, we show more visualized results on NYUDv2 and BIPED, which are shown in Fig. 5 and Fig. 6, respectively. These visualized results demonstrate that the EDSNN can generate clean and refined edge maps, which are consistent with that of BSDS500.

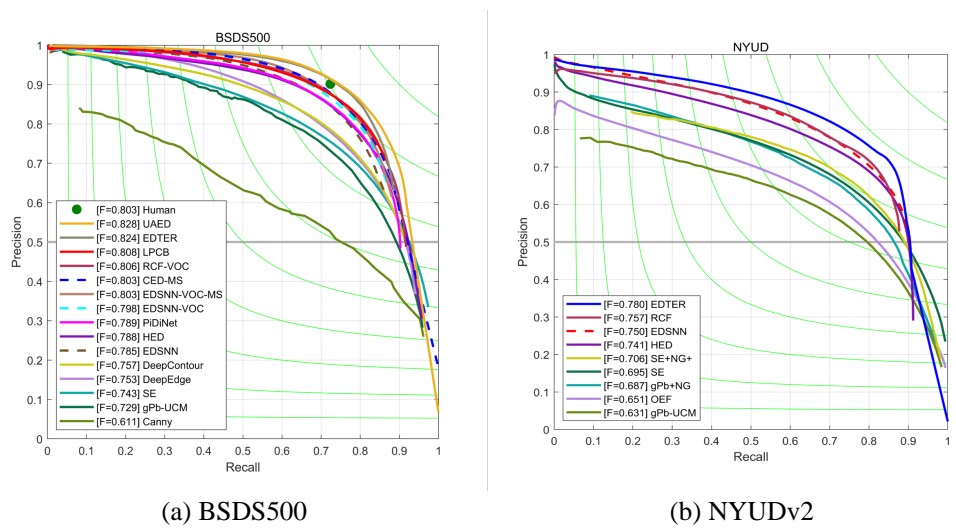

Figure 4: Precision-Recall curves on BSDS500 and NYUDv2, respectively.

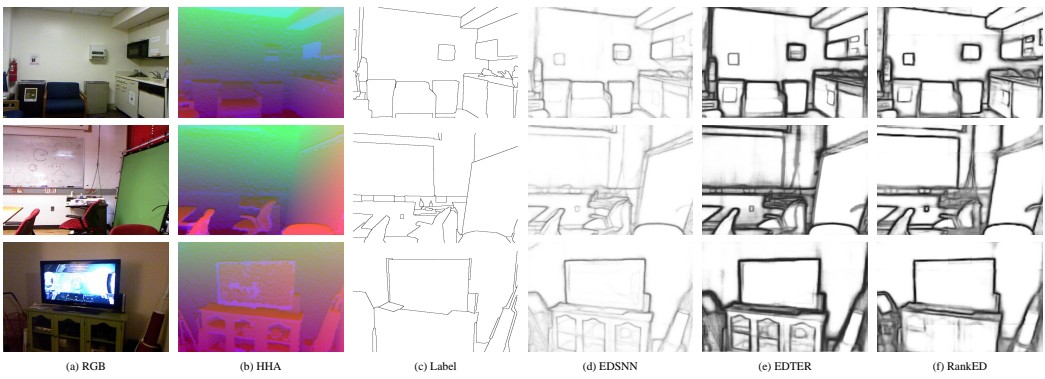

Figure 5: Some examples from different SOTA methods on NYUDv2.

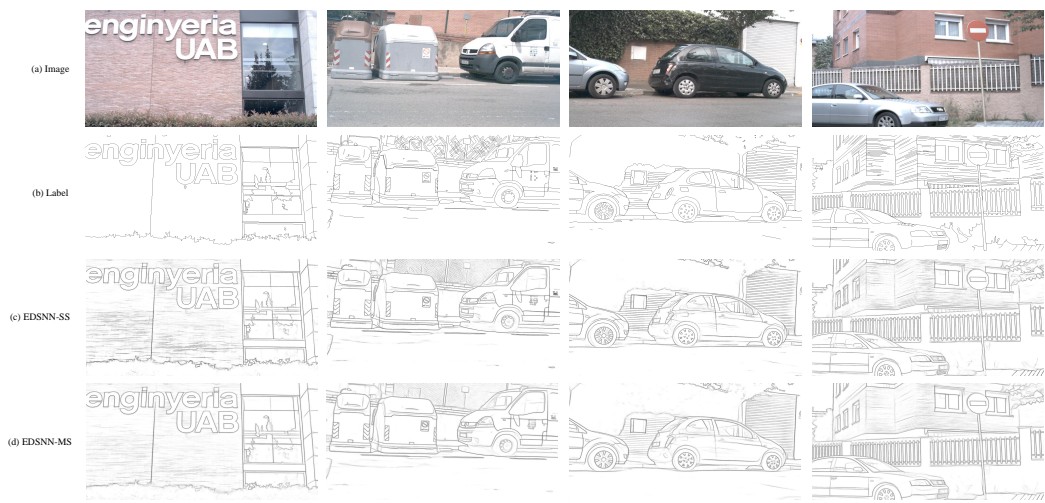

Figure 6: More examples from EDSNN on BIPED. EDSNN-SS indicates the predictions with single-scale testing, and EDSNN-MS indicates the predictions with multi-scale testing.

