# OpenReview forum: "EDSNN: Edge Detection with Spiking Neuron Network"
_ICLR.cc/2025/Conference — ICLR 2025 Conference Withdrawn Submission_

### Official Review · Reviewer_aG8h · 2024-10-25

**Soundness:** 2
**Presentation:** 3
**Contribution:** 1
**Rating:** 3
**Confidence:** 4

**Summary:**

In this paper, an SNN-based edge detection method is proposed to achieve lower energy consumption, with an encoder-decoder network architecture. This work constructs a Spiking Multi-Scale Block (SMSB) to utilize multi-scale information and present a Membrane Average Decoding (MAD) method in the prediction block. Though achieving theoretical low energy consumption, this work looks like an incremental application of SNN in a specific task, limiting its novelty and contributions.

**Strengths:**

1. The paper is well-organized and easy to follow.

2. In theory, the proposed method achieves low energy consumption.

**Weaknesses:**

1. The authors claim that this work proposes the first SNN for edge detection. However, it seems that there are some existing SNN works for edge detection, such as those in the following papers.
[1] Segmentation and edge detection based on spiking neural network model.
[2] GPU implementation of spiking neural networks for edge detection.
[3] Emulating spiking neural networks for edge detection on FPGA hardware.

2. This work looks like an application of SNN in edge detection, with limited novelty. As many works have tried to employ SNN in some vision tasks, the contribution and inspiration to the community are also limited. The paper proposes many designs for edge detection, such as multi-scale blocks (which look like U-Net). However, similar designs have been used in many other vision tasks. The proposed method lacks efficient designs for the specific vision task, i.e., edge detection.

3. The theoretical low energy consumption comes from SNN itself. More proof should be given to verify that the proposed designs contribute to the low energy consumption.

4. The paper does not explain the symbols in the equations enough, which may cause misunderstanding.

5. In many cases, the bold value means the best performance in a certain metric. The paper doesn't follow it. Thus explanations for the bold values are expected in the caption of the tables to avoid misunderstanding.

6. Equation (3) and Equation (4) can be improved. From a mathematical point of view, this is not a rigorous way of expression.

**Questions:**

According to theoretical energy values, the main claimed advantage of the work is low energy consumption. How do you implement it in practice? What kind of hardware do you employ to achieve the theoretical energy values? Maybe more details are required.

---

### Official Review · Reviewer_Uk8z · 2024-10-31

**Soundness:** 2
**Presentation:** 2
**Contribution:** 2
**Rating:** 3
**Confidence:** 4

**Summary:**

In this work, the authors propose the first SNN-based method for edge detection, named EDSNN (Edge Detection with Spiking Neural Networks). Their approach uses a Spiking Multi-Scale Block (SMSB) to generate precise and clean edge maps.

**Strengths:**

The paper is easy to follow.

It would be helpful to clarify the specific advantages SNNs offer for edge detection applications.

**Weaknesses:**

The main challenges in edge detection are precise edge localization and generating clean edge maps. However, the connection between these challenges and the use of SNNs remains unclear. It would strengthen the paper to explain why SNNs are particularly suited to address these issues.

Multi-Scale Block (MSB) is a generalizable technique and could also be implemented with binarized neural networks (BNNs). Since BNNs don’t rely on neuromorphic hardware, the proposed method might not need to wait for neuromorphic hardware advancements to be useful.

When the time step (T) equals 1, the SNN behaves similarly to a BNN. The authors list these results in the table, but it may weaken the case for SNN-specific benefits.

The paper could improve citation consistency. For example, "Maass (1997)" is appropriate at the beginning of a sentence, while "(Maass, 1997)" fits better at the end. Consistent formatting would improve readability.

**Questions:**

Please see the points under Disadvantages—specifically, how does SNN usage uniquely benefit edge detection compared to other neural network approaches?

---

### Official Review · Reviewer_nhvG · 2024-11-03

**Soundness:** 2
**Presentation:** 3
**Contribution:** 3
**Rating:** 5
**Confidence:** 3

**Summary:**

This paper is the first to use SNN for edge detection tasks. The paper proposes the SMSB method to improve the model's ability to capture multi-scale information. Furthermore, the MAD method is used to decode and improve the expressiveness of the model. The author tested it on multiple edge detection tasks. It demonstrates the potential of SNN in edge detection and improves the feasible downstream tasks of SNN in the field of computer vision.

**Strengths:**

1. This paper first proposed the SNN solution for edge detection.
2. The proposed method is relatively effective. SMSB is a reasonable approach. Furthermore, the article provides visualization, and EDSNN shows a good model performance.

**Weaknesses:**

1. Has the multi-scale convolution method appeared in other fields? Because this method of using multi-scale convolution is very trivial, I am concerned about the lack of innovation in your Edge Detection.
2. Can other SNN infrastructures complete edge detection tasks? Is there a comparison between EDSNN and other architectures in edge detection (to reflect the effectiveness of your modifications to the SNN architecture).

**Questions:**

1. Please explain the innovation of the SMSB method. There may be two effective ways to do this: a. Compare the differences between the SMSB you proposed and other multi-scale convolutions. b. Give the uniqueness of the application of SMSB in SNN. Is there any special adaptation for SNN? Perhaps you can analyze it.
2. Will this multi-scale operation lead to excessive computational overhead? In other words, if the same computational overhead is used, will other methods also have better results? Is this performance improvement necessarily brought about by multi-scale?

---

### Official Review · Reviewer_aqbX · 2024-11-04

**Soundness:** 2
**Presentation:** 3
**Contribution:** 2
**Rating:** 5
**Confidence:** 4

**Summary:**

This work proposes EDSNN, the first spiking neural network model for object edge detection tasks. By leveraging the energy-efficient nature of SNN and applying novel methods such as Spiking Multi-Scale Block (SMSB) and Membrane Average Decoding (MAD), the model significantly reduces energy consumption during training in comparison with ANNs while maintaining a comparable performance.

**Strengths:**

1. The paper is well-organized, and the authors use numerous figures to clearly illustrate the model structure and present sample results effectively.

2. It is the first spiking neural network model for edge detection and the authors demonstrate the proposed model’s performance across a variety of experiments on different datasets, showing that it achieves comparable results while significantly reducing energy consumption.

3. The authors also conduct comprehensive ablation studies to highlight the impact of each component within the proposed method.

**Weaknesses:**

1. As shown in Table 1, replacing SMSB with a standard 3×3 convolution in the decoder results in only a minor drop of 0.004 in ODS, 0.006 in OIS, and 0.003 in AP. I question whether it is accurate to describe this as a **significant drop**. A similar observation applies to the ablation study on the decoding method in Table 1. While I acknowledge the energy savings achieved by MAD, the accuracy improvement appears minimal.

2. In SMSB, the authors introduce five convolutional layers and aggregate them to provide multi-scale features. It would be beneficial to visualize the activation maps from these convolutional layers to demonstrate their practical effectiveness.

3. The novelty of the approach is limited. The proposed EDSNN architecture and SMSB bear strong similarities to the architecture and second-order derivative-based multi-scale contextual enhancement module (SDMCM) in [1]. The authors have essentially provided a simplified version, substituting the activation function from the original ANN model with an IF node.

[1] Changsong Liu et.al, "Learning to utilize image second-order derivative information for crisp edge detection"

**Questions:**

Is there a typo in Table 1? The energy consumption (7.56) in the second row of the last block differs from other rows with the same configuration (7.59), even though the other metrics are identical.

---

### Official Review · Reviewer_jVdo · 2024-11-04

**Soundness:** 3
**Presentation:** 2
**Contribution:** 2
**Rating:** 3
**Confidence:** 4

**Summary:**

In this work, the authors proposed an edge detection model for spiking neural networks (SNNs) called EDSNN. The model is based on UNet architecture (encoding-decoding approach) with spiking multi-scale block (SMSB) and membrane average decoding (MAD). The authors evaluated the proposed model on various datasets, such as BSDS500, BYUDv2, and BIPED.

**Strengths:**

This study is an initial work utilizing SNN to perform edge detection tasks with low power. The proposed model shows inference results comparable to those of DNN with low power consumption.

**Weaknesses:**

Despite the high inference performance of this study, several concerns should be resolved before this paper can be published.

- Lack of novelty
    - SMSB - The model for multi-scale features was used in the existing edge detection model, PiDINet. What are the differences from the existing methods?
    - In terms of model architecture what are the differences between the following study that utilized UNet architecture for edge detection? (HED-UNet: Combined segmentation and edge detection for monitoring the Antarctic coastline)
    - The Spiking VGG network and Membrane Average Decoding proposed in this paper have already been widely used in various studies related to deep SNN. What are the differences between the proposed method and those studies?

- The proposed method is very similar to the existing DNN method. What was the biggest difficulty in performing edge detection with deep SNN, and how did the authors overcome it?

- Detailed descriptions of the model architecture should be added. How is the Spiking-VGG Block and Up Pred Block in Figure 1 structured? Did you experiment with various datasets for the same model?

- Miscellaneous
    - There are duplicate definitions of abbreviations.

**Questions:**

- How does the energy consumption compare to DNNs?
- How does the precision-recall curve of the proposed method compare with other studies?

---

### Note · Authors · 2024-11-13

I have read and agree with the venue's withdrawal policy on behalf of myself and my co-authors.